# Assessment of Tendon Prestressing after Long-Term Service via the Barkhausen Noise Technique

**DOI:** 10.3390/ma12203450

**Published:** 2019-10-22

**Authors:** Miroslav Neslušan, František Bahleda, Martin Moravčík, Katarína Zgútová, Filip Pastorek

**Affiliations:** 1Faculty of Mechanical Engineering, University of Žilina, Univerzitná 1, 01026 Žilina, Slovakia; 2Faculty of Civil Engineering, University of Žilina, Univerzitná 1, 01026 Žilina, Slovakia; frantisek.bahleda@fstav.uniza.sk (F.B.); martin.moravcik@fstav.uniza.sk (M.M.); katarina.zgutova@fstav.uniza.sk (K.Z.); 3Research Centre, University of Žilina, Univerzitná 1, 01026 Žilina, Slovakia; filip.pastorek@rc.uniza.sk

**Keywords:** Barkhausen noise, prestressed tendon, bridge

## Abstract

This paper deals with the assessment of a real prestressed tendon by the use of Barkhausen noise emission. The tendon was obtained from a real highway bridge after 33 years in service. Barkhausen noise is studied as a function of the stress state, and the Barkhausen noise signals received directly from the tendon on the real bridge are compared with the Barkhausen noise signals received from the tendon during loading in the laboratory. Assessment of the prestressing is based on the analysis of the effective value of the Barkhausen noise signal as well as the position in which the Barkhausen noise envelopes attain a maximum.

## 1. Introduction

Tendons are very important, considering the reliable and long-term service of bridges, as the most important load-bearing components. The acceptable long-term shape stability, resistance against the variable loading modes, together with superimposing resistance against the aggressive environmental effects of bridges are possible, since these constructions represent the typical composite structure in which concrete provides corrosion protection to wires and/or tendons as the main bearing elements. In order to secure the required functional properties of bridges, the tendons and the corresponding wires are prestressed during bridge installation. However, the long-term service of bridges can remarkably alter this prestressing.

Two different ways in which the tendon can be altered should be mentioned. The tendon can increase due to the reduction of the cross-sectional area of a wire as a result of its corrosion, whereas the prestressing can decrease due to its relaxation as a result of long-term service. The real prestressing in the tendon is usually driven by the superimposing influence of both effects. As soon as a wire stress exceeds the critical threshold (yield strength), dislocation slip occurs (together with an irreversible elongation of the wire) as a result of the plastic regime loading. Xu and Chen [1] carried out tensile tests of corroded tendons and found that the yield strength, as well as the ultimate strength, remarkably decreased with a progressive increase of the corrosion extent. Furthermore, rupture failure can occur despite the exerted stress being less than the yield strength of the steel matrix [2,3,4]. Li et al. [2] found that stress corrosion in one wire redistributes the load within the other wires which, in turn, contributes to the subsequent overloading of the non-corroded wires. 

On the other hand, decreased wire prestressing negatively affects the bearing capacity of a bridge. Certain alterations of the tendons have to be considered when a bridge is in service over decades. However, these alterations should not exceed the acceptable range and a non-destructive technique, adapted for this purpose, would be beneficial. The corrosion extent and superimposed prestressing of the tendon have been analysed in some studies. Anania et al. [5] reported that the collapse of the Petrulla viaduct was due to poor construction practice and accelerated tendon corrosion, which consequently reduced the tendon’s cross-section. Singh et al. [6] attributed the failure of the tendons as a result of mechanical wear, stress overload, as well as corrosion pitting. 

Therefore, many techniques have been developed for the detection and reliable assessment of the rope wire state expressed in the aforementioned aspects. Li et al. [2] developed a cluster analysis of stress corrosion mechanisms by the use of acoustic emission particle swarm optimisation. Martin et al. [7] applied ultrasonic tomography, Peng and Wang [8] used gamma rays, whereas Christen et al. [9] applied a magnetic flux leakage method for the localisation of defects in stay cables. Moreover, electromagnetic testing is very often adapted for such purposes [10].

Liu et al. [11] reported that ultrasonic wave propagation is a promising technique for non-destructive testing of multi-wire structures. Zhang et al. [12] presented a quantitative investigation on the longitudinal variability in the cross-sectional areas of seven pre-stressed wires. Xu et al. [13] proposed an analytical model of damaged multi-wire cables with broken wires. Panfilov and Pischulev et al. [14] presented the calculations of a monolithic slab and beam floor with a prestressed wire rope reinforcement. Musikhin [15] reported on a model based on the Kirchhoff differential equations for the determination of the real stress-strain state of a steel wire rope elements. Filho et al. [16] reported about the effect of hydrogen release at room temperature on the ductility of steel wire rods for prestressed concrete. 

This paper discusses the potential of magnetic Barkhausen noise (MBN) for the assessment of the real prestressing in steel wires after their long-term service. MBN occurs in ferromagnetic structures during cyclic magnetisation (or cyclic mechanical loading), and MBN pulses can be received on the free surface due to discontinuous domain walls’ motion [17]. Various lattice defects pin domain walls and their irreversible motion occurs as soon as the magnetic field attains a critical level, equal to the pinning strength of the pinning sites. MBN is a function of the pinning sites and strongly affects the free path of the domain walls’ motion and, thus, the MBN magnitude. Domain walls are pinned by dislocation tangles, precipitates, grain boundaries, non-ferromagnetic particles, or phases, etc. [18,19,20,21]. On the other hand, the stress state strongly affects the domain walls’ alignment. It is well known that iron alloys usually exhibit positive magnetostriction. For this reason, compressive stresses tend to lower the magnitude of MBN pulses, whereas tensile stresses increase the pulse magnitude [22,23,24]. Hence, the MBN technique can be employed for the assessment of stresses in ferromagnetic bodies.

The MBN technique is very fast, portable and sensitive against surface damage, expressed in such terms as the alteration of the microstructure as well as the stress state. The MBN technique has already been adapted for assessment of the corrosion extent in wires [25]. For these reasons, this study investigates the potential of the MBN technique for assessment of the prestressing in tendons.

## 2. Experimental Conditions

Experiments were carried out on seven wire tendons containing six outer wires of nominal diameter 4.6 mm, and corresponding nominal cross-sectional area *A*_0_ = 16.62 mm^2^, as well as the central wire of nominal diameter of 5 mm (see Figure 1). MBN measurements were carried out on only five outer wires (one outer wire as well as the central wire could not be measured directly on the bridge, the measured wires were numbered as wire n.1, wire n.2, etc.). The wires, with approximate tensile strength 1800 MPa and elastic modulus of 190 GPa, were subjected to 33 years of operation in the Liptovský Hrádok highway bridge (Northern Slovakia). The initial prestress in the wires was about 1200 MPa. The chemical composition of the investigated wires is shown in Table 1.

This bridge was subjected to demolition due to excessive deformation during loading as well as decomposition of some of the concrete blocks. Hence, the prestressed components could be removed (cut off) from the bridge. However, in the initial phase, it was necessary to measure MBN signals before the cables were removed. For this reason, an approximately 850 mm long section was prepared for measurement. Concrete was gently removed from the rope surface and the rope surface was cleaned by the use of a wire brush. As soon as the MBN signals on the pre-stressed rope were picked-up, this section was cut off. Subsequently, the rope was loaded in the laboratory by using a Matest H011P105 machine (Matest S.p.A., Bergamo, Italy) on the predefined tensile stresses, and the corresponding MBN measurements were also carried out. Finally, the MBN signals (extracted MBN parameters and their evolution) were compared with the signals picked-up directly on the bridge.

MBN was measured by the use of a RollScan 350 (Stresstech, Jyväskylä, Finland) and analysed with MicroScan 600 software (magnetising voltage: 5 V; magnetising frequency: 125 Hz; sensor type: S1-18-12-01; frequency range of MBN pulses in the range of 10–1000 kHz, (Stresstech, Jyväskylä, Finland). MBN values were obtained by averaging 10 MBN bursts (five magnetising cycles). MBN refers to the RMS (root mean square-effective) value of the signal. The magnetisation of the wires was carried out along the longitudinal axis (see Figure 1). In addition to the conventional MBN parameter (RMS value of the signal), the peak position (PP) of the MBN envelope was also analysed. The PP of the MBN usually refers to the position of the magnetic field in which the MBN envelope attains a maximum. The field between the magnetising poles was measured by the use of a Hall probe in the preliminary phase of the experiments for the magnetising conditions indicated in the study. The strength of the magnetising field was measured in mT and converted into kA·m^−1^ afterward.

To observe the surface state of the analysed wires (by the use of scanning electron microscopy (SEM, Tescan Brno s.r.o., Brno, Czech Reublic) as well as light microscopy (Nikon Instruments Inc., New York, NY, USA) in the direction of loading (longitudinal direction), as well as in the perpendicular direction (especially with respect of their corrosion as well as the preferential orientation of the matrix), the wires were cut by the use of a Struers Secotom-50 and routinely prepared for metallographic observations: hot moulded, ground, polished and etched by 3% Nital for 10 s.

Vickers microhardness (HV 0.1) testing was conducted using an Innova Test 400^TM^ tester (Innovatest, Maastricht, the Netherlands) by applying a force of 100*g* for 10 s. The microhardness was determined by averaging three repeat measurements at a distance of approximately 30 μm from the free surface.

## 3. Results and Discussion

The microstructure of the wires is entirely composed of ferrite containing fine dispersed globular cementite known as sorbite. Cementite is a very hard ferromagnetic phase [26]. Due to this, the domain walls are very difficult to unpin under the magnetic field produced by the employed sensor and, therefore, the entire MBN emission from the wires originates from the ferrite only. 

Metallographic images, as depicted in Figure 2 and Figure 3, illustrate that the tendon surface exhibits certain corrosion damage of a variable extent as well as decarburised regions. The corrosion extent is represented by small or larger dimples (Figure 3b) on the surface in which the corrosion products were settled. These corroded particles were removed from the wire’s surface by the use of a wire brush during the cleaning of the tendons before MBN measurement. Decarburised regions on the wire’s surface appear white in the metallographic images and the microhardness HV0.1 at a depth of approximately 25 μm drops to 365 ± 13, as compared with the nearly untouched regions (such as that illustrated in Figure 2a) in which the microhardness HV0.1 is 454 ± 14. Decarburisation is a thermally initiated process which depletes the carbon content which, in turn, plays the major role when considering the decreased hardness [27]. It is estimated that the decarburisation of the wires, which occurs randomly on their surface, was initiated by the wires’ tempering in the final phase of heat treatment. These figures illustrate that the corrosion extent, as well as the degree of decarburisation, varies in the different positions (with respect to the cross-sectional area—Figure 3, as well as in the longitudinal direction—Figure 2). 

As is well known and reported in a previous study [25], the corrosion process on the wire surface is very complex. Corrosion is an electrochemical process. During this electrochemical process, localised areas of the exposed surface become relatively anodic or cathodic with respect to one another. Corrosion products, which are produced in some regions, have a different chemical composition as compared with the original matrix, which leads to changes in the anodic and cathodic areas. For this reason, previously uncorroded areas are attacked and corroded. Factors such as variations in the composition and/or microstructure of the steel, the presence of impurities, uneven internal stress, and/or exposure to a non-uniform environment, all affect the corrosion process, especially the non-homogenous distribution of the corrosion extent. 

This is a reason why the regions of different corrosion extent can be found on the same wire and/or the same cross-section (as Figure 2 and Figure 3 illustrate). Figure 2 and Figure 3 illustrate the corrosion extent as well as decarburised regions for the different regions of wire n.2 only. A similar character of the surfaces for the other four wires was observed in which there were regions nearly untouched by corrosion and/or decarburisation, neighbouring those where this had occurred to a higher extent. MBN signals obtained from the wire’s surface represent the integrated value over the certain areas in which nearly unaffected (expressed in term corrosion and/or decarburisation) regions are mixed with those containing corrosion dimples or/and white zones.

Figure 4, Figure 5, Figure 6, Figure 7 and Figure 8 illustrate the evolution of MBN (its RMS, effective value) and *PP* for five different wires of the rope as a function of applied stress. It can be seen that the MBN values increase along with increasing tensile stresses. However, as soon as the tensile stress attains a certain maximum, the MBN drops for higher stresses. On the other hand, the evolution of *PP* exhibits the opposite trend. The *PP* values at lower stresses decrease followed by a visible increase at the end of the test when higher stresses are exerted. In can be found that the evolution of *PP* is more or less inversely proportional to the evolution of MBN.

The values of MBN as well as *PP*, measured directly on the bridge, are indicated by the red line for MBN and the black line for *PP* in each figure. As a result of the non-monotonous evolution of MBN (respective *PP*) it can be found that in many cases these lines intersect the evolution of MBN (respective *PP*) at two different points. For this reason, the values measured on the bridge can be associated with two different stress levels. 

On the basis of the visual inspection of the tendons it was found that neither of the measured wires suffered from a heavy corrosion extent. Wire surfaces (especially in the positions in which the MBN was measured) exhibited no corrosion evidence and no reduction of cross-sectional area. Surface corrosion was evidenced by the metallographic images only. Therefore, it is considered that the pre-stress in all analysed wires should be comparable. Furthermore, it is considered that the pre-stress in a wire obtained by the use of MBN and *PP*, should be comparable as well. The intersection points of lines with MBN and *PP* evolutions in Figure 4, Figure 5, Figure 6, Figure 7 and Figure 8 give different prestresses (when the lines intersect the evolutions at two different points). For this reason, the diagram (Figure 9) in which prestressing is determined by the use of MBN versus *PP*, can be plotted. Such a diagram should contain 10 different points (two intersections for five different wires). However, only eight points can be found in Figure 9 since only one intersection point can be found (for this reason one or two coordinates are missing) in some cases (for instance Figure 6). Figure 9 shows that two points for the lower tensile stresses are missing. Moreover, points originating from the higher stress levels lie much closer to the red line which indicates how close the prestressed values obtained from the MBN and *PP* parameters are.

Taking into consideration the aforementioned aspects, it can be found that the true prestressing in the analysed wires is in the range from 1000 to 1180 MPa. To prove such a finding, a further insight into the MBN can be carried out by the use of the maxima, shape and position (with respect to the magnetic field strength) of the MBN envelopes. Considering the comparable MBN and *PP* obtained from the laboratory tests with those measured on the prestressed bridge, the MBN envelopes should overlap much more than those in which the MBN and *PP* differ. Figure 10 shows the MBN envelopes for the different stress levels as well as the MBN envelope extracted from the raw MBN signal measured directly on the prestressed rope. It can be clearly seen that the MBN envelope for 250 MPa exhibits a higher maximum (equal to the higher MBN) than the prestressed on the bridge, whereas the MBN envelope for 50 MPa is shifted to higher magnetic fields (equal to the higher *PP*). Only the envelope for 1000 MPa coincides with the envelope measured directly on the bridge for wire n.2. 

## 4. Discussion of the Obtained Results

It has already been reported that corrosion on a wire surface attenuates the MBN emission [25]. On the one hand, it was determined that corrosion products were removed from the surface, whereas, on the other hand, dimples on the surface increase the surface roughness which, in turn, contributes to the attenuation of MBN pulses propagating towards the pick-up coil [28]. However, this effect can be compensated for by the presence of a decarburised region which produces a higher MBN as compared with an unaffected region [26]. For this reason, MBN is driven by the superimposing effects counteracting each other. Comparing wire to wire, the MBN and *PP* evolutions are more or less modified. MBN (*PP*) values, maxima (minima) of the evolutions, as well as the positions in which they can be found, vary more or less as a result of the non-homogenous and random distribution of corrosion and/or decarburisation (their variable extent), when the MBN pick-up coil integrates the signal over the different regions, etc. The redistribution of the exerted load during decades of long-term service should also be considered [4]. Furthermore, MBN is very sensitive against changes of microstructure and any alterations during the wire’s production, and/or their thermomechanical treatment, would contribute to the difference in MBN (*PP*) and their evolution against stresses. 

It is also worth mentioning that MBN for Fe alloys tends to increase along with increasing tensile stresses in the elastic region of loading [24,29,30] due to positive magnetostriction. However, the evolutions of MBN in Figure 4, Figure 5, Figure 6, Figure 7 and Figure 8 also exhibit the descending region, especially for higher tensile stresses. Such behaviour is typical for Ni-based alloys due to quite high negative magnetostriction [31,32]. Behaviour such as that depicted by Figure 4, Figure 5, Figure 6, Figure 7 and Figure 8 has been already reported and explained by Amiri et al. [17,33]. As the authors reported, there is a competition between two anisotropies to determine the easy axis of magnetisation under applied stress. At low applied stresses, crystal anisotropy plays the main role in the magnetisation process and MBN increases while, at higher stresses, stress anisotropy (magnetoelasticity) plays the major role and MBN decreases. This means that the easy axis is controlled by crystal anisotropy in the case of the lower tensile stresses, thus, domains, and the corresponding domain walls, turn into the direction of the magnetic easy axis. On the other hand, at higher tensile stresses, the domains and the corresponding domain walls are forced to turn into the direction of the new easy axis which is controlled by stress [17,33,34]. Evolution of MBN versus the stress state is usually driven by the competition between magnetocrystalline and magnetoelastic energies. Magnetocrystalline energy *E_a_* strongly depends on the magnetocrystalline anisotropy as Equation (1) indicates for cubic crystals [32]:(1)$$E_{a}=K_{1}\left(\alpha_{1}^{2}\alpha_{2}^{2}+\alpha_{2}^{2}\alpha_{3}^{2}+\alpha_{1}^{2}\alpha_{3}^{2}\right)+K_{2}\left(\alpha_{1}^{2}\alpha_{2}^{2}\alpha_{3}^{2}\right)+K_{3}\left(\alpha_{1}^{2}\alpha_{2}^{2}+\alpha_{2}^{2}\alpha_{3}^{2}+\alpha_{1}^{2}\alpha_{3}^{2}\right)+\ldots$$
where *α*_1_, *α*_2_ and *α*_3_ are the direction cosines of the magnetisation vector with respect to three cube edges, and *K*_1_, *K*_2_ and *K*_3_ represent the magnetocrystalline anisotropy constants. 

On the other hand, the magnetoelastic energy *E_σ_* can be expressed as follows [32]:(2)$$E_{\sigma }=-\frac{3}{2}\lambda_{s}cos^{2}\varphi$$
where *λ_s_* is the isotropic magnetostriction and *φ* defines the angle between the direction of magnetisation and the direction of the exerted stress *σ*.

It should be noticed that the magnetocrystalline energy can be strongly consumed during the cold forming of wires in dies (during manufacturing). Such processes produce a matrix exhibiting the crystallographic texture similar to that illustrated in Figure 2 and Figure 11, and the corresponding, quite large, uniaxial anisotropy [32] with its easy axis in the longitudinal direction (the direction of exerted tensile stress). For this reason, the increasing tensile stresses make the magnetocrystalline and magnetoelastic energies equal, quite early, and further increase in tensile stresses lowers the MBN emission in the tendons and increases the *PP*.

## 5. Conclusions

Assessment of prestressing tendons after long-term service is very important for the evaluation of the estimated bridge lifetime. Non-destructive testing is of vital importance since any removal of the ropes is unacceptable as this would lower the bearing capacity of the bridge. Assessment of the tendon prestressing by the use of MBN in this study is based on a comparison of the MBN measured directly on the bridge and in the laboratory, using the tendons cut from the bridge. Due to the announced demolition of the investigated bridge, cutting off one rope was acceptable, and the experience obtained during subsequent analysis could be considered as a training period which provides data for further possible applications. It is considered that the assessment of the prestressing in tendons in real bridges, subjected to further service, should be carried out in semi-destructive or fully non-destructive manners. The non-destructive manner means that the information (data) obtained during the training period could be employed for the assessment of prestressing in tendons based on MBN measurements on the bridge only. The semi-destructive assessment considers that the MBN measurement on a prestressed tendon would be carried out for multiple wires and only one wire would need to be cut off (on the basis of a previous communication with the people responsible for the bridge in service, it would be acceptable), since it is considered (on the basis of Figure 9) that the prestresses in the different wires are more or less comparable and fluctuate in quite a narrow range. 

## Figures and Tables

**Figure 1 materials-12-03450-f001:**
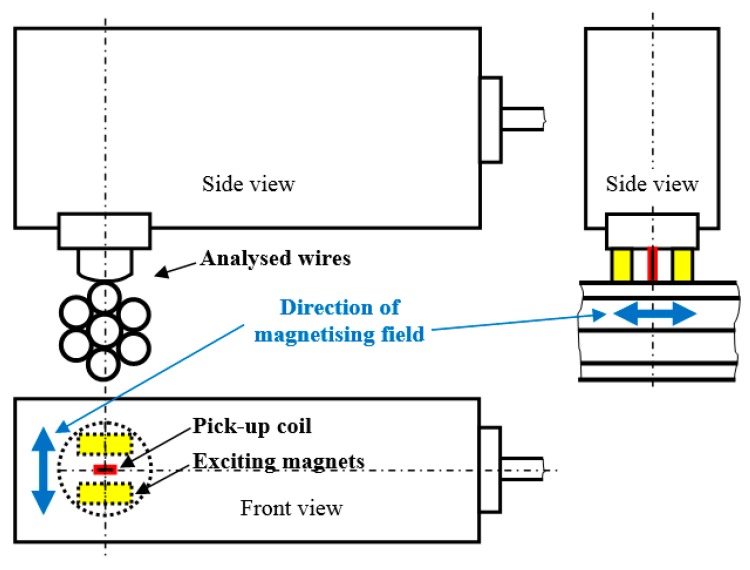
Brief sketch of the MBN measurement.

**Figure 2 materials-12-03450-f002:**
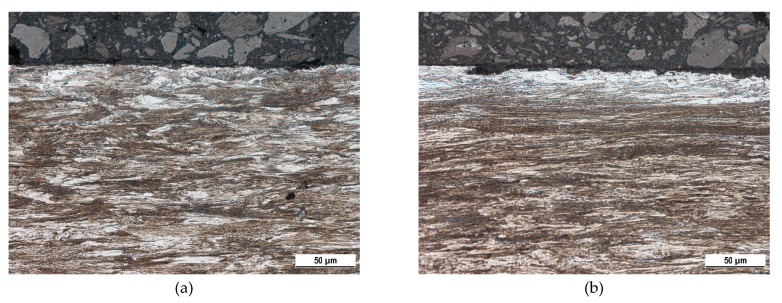
Metallographic images of wire n.2’s longitudinal direction. (**a**) Region 1; (**b**) region 2.

**Figure 3 materials-12-03450-f003:**
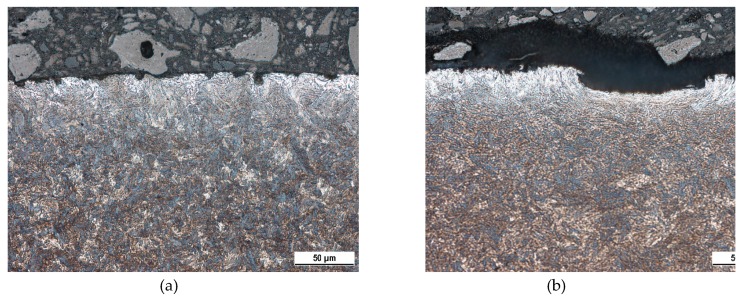
Metallographic images of wire n.2’s cross-sectional view. (**a**) Region 1; (**b**) region 2.

**Figure 4 materials-12-03450-f004:**
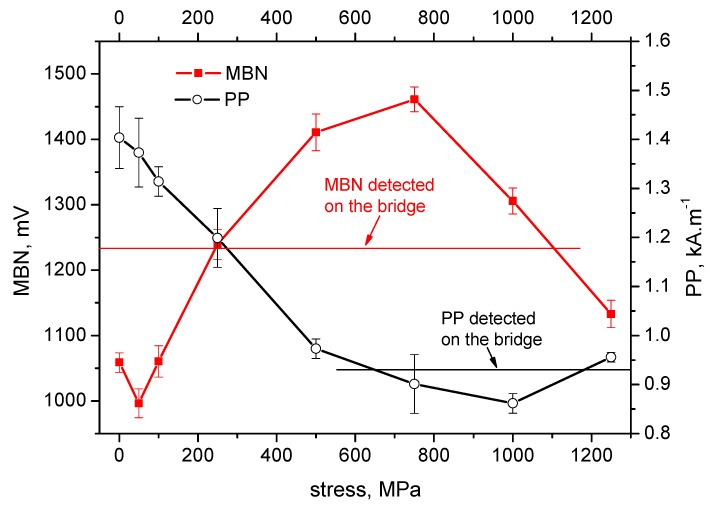
MBN and *PP* as a function of stress—wire n.1.

**Figure 5 materials-12-03450-f005:**
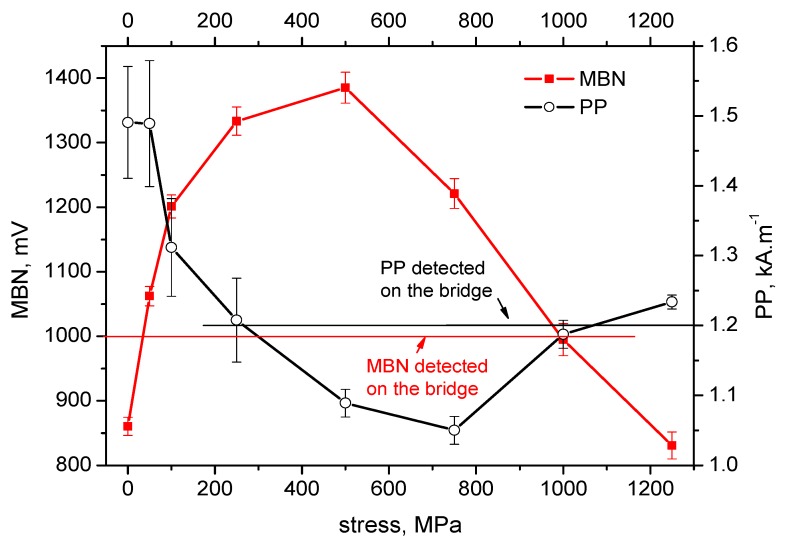
MBN and *PP* as a function of stress—wire n.2.

**Figure 6 materials-12-03450-f006:**
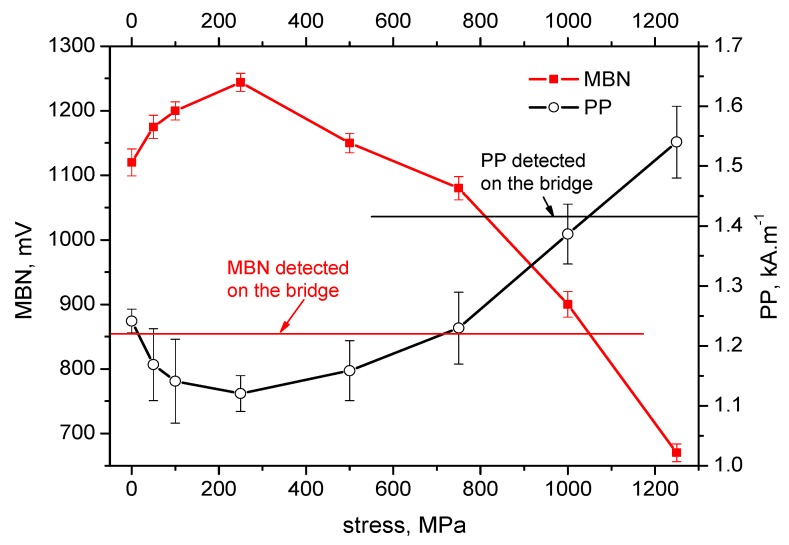
MBN and *PP* as a function of stress—wire n.3.

**Figure 7 materials-12-03450-f007:**
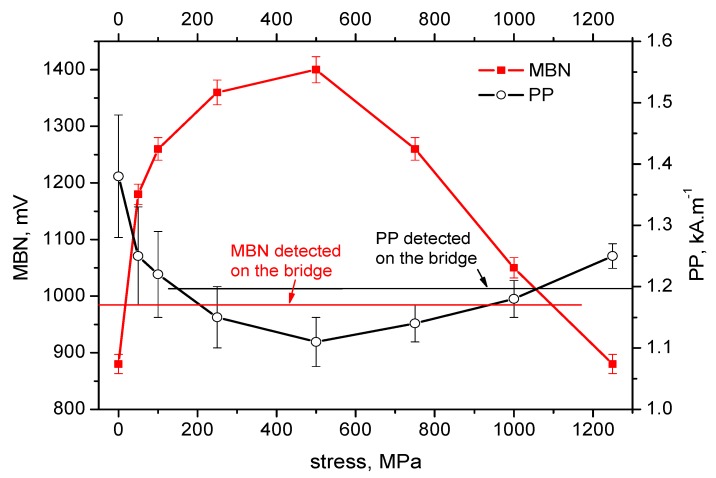
MBN and *PP* as a function of stress—wire n.4.

**Figure 8 materials-12-03450-f008:**
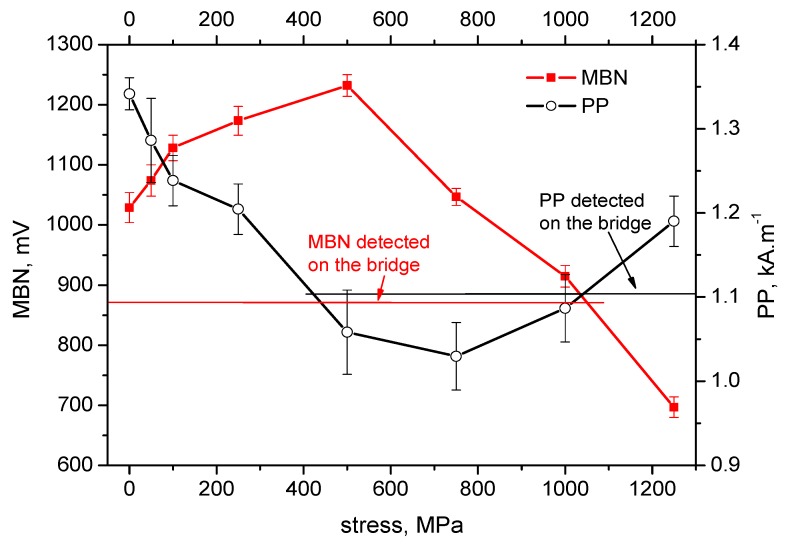
MBN and *PP* as a function of stress—wire n.5.

**Figure 9 materials-12-03450-f009:**
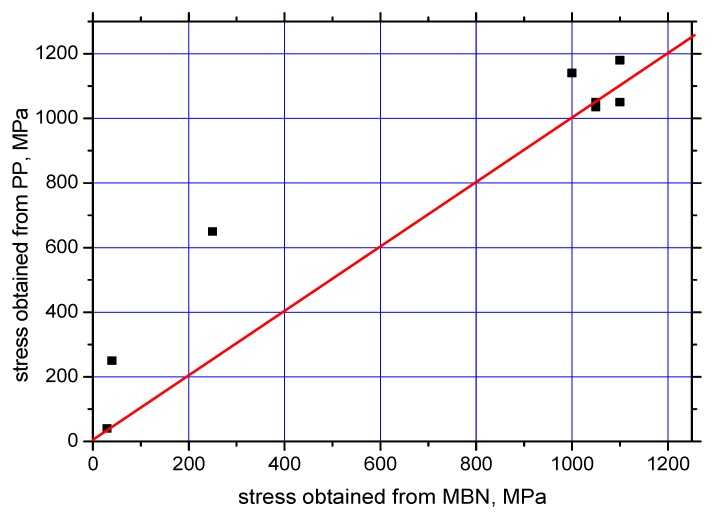
Prestress obtained from MBN versus *PP*.

**Figure 10 materials-12-03450-f010:**
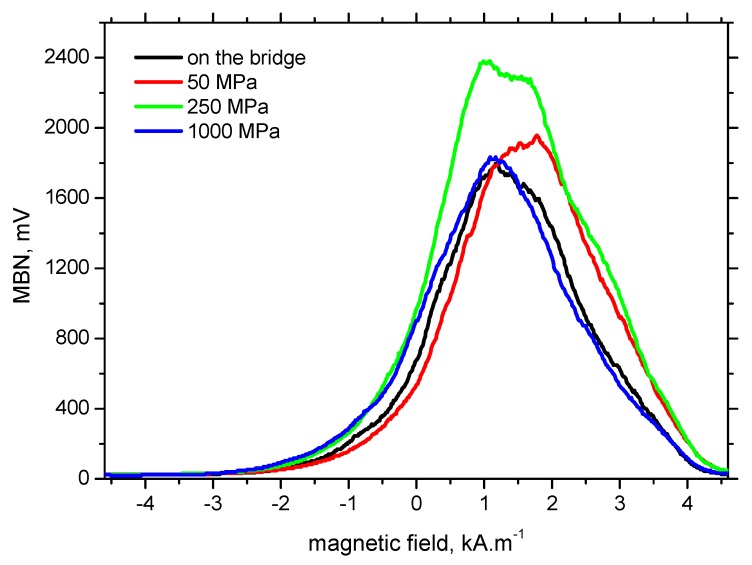
MBN envelopes for wire n.2.

**Figure 11 materials-12-03450-f011:**
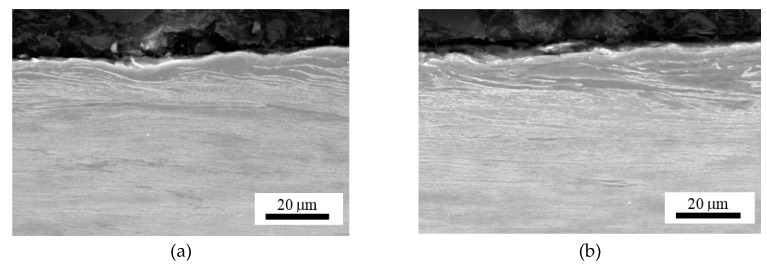
SEM images of preferential orientation of rope wire n.2. (**a**) Region 1; (**b**) region 2.

**Table 1 materials-12-03450-t001:** Chemical composition of the investigated wires in wt%.

Fe	C	Mn	Si	P	S	Cr	Ni	Cu	V	Mo
balance	1.060	0.361	0.408	0.029	0.068	0.128	0.048	0.184	0.005	<0.001

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
