# Peer review of "Assessment of Tendon Prestressing after Long-Term Service via the Barkhausen Noise Technique"

_materials, 2019, doi:10.3390/ma12203450_

Round 1

Reviewer 1 Report

Do you measure the initial rope in operation (inside the bridge) only in one spatial point or along the length in different points?

Figs. 4-8 could be jointed in two figures: 4(a) presents all rms dependence and 4(b) presents all PP dependence. It could be problem to show all operation (bridge) values in one graph, but it could look better.

How do you evaluate the field in kA/m for Rollscan measurements?

I do not like much the discussion about two anisotropy axes with the stress. There are more classical older works proposing dislocation formation as a reason for the following rms decrease. 

Author Response

Reviewer: Do you measure the initial rope in operation (inside the bridge) only in one spatial point or along the length in different points?

Response: Yes, we measured MBN emission in one point along the length on the 5 different wires only. Please consider that the wires are inside the steel tube. This tube is filled by concrete and the tube is inside of the bridge construction. Being so, it is necessary to remove certain volume of concrete to find the steel tube. The tube made of steel should be gently removed (cut off) and the concrete inside of the tube should be also gently removed. Finally the surface of the wires of tendon should be also cleaned as it is indicated in the study. The aforementioned text indicate that the measurement of the MBN on the wires is easy and fast but removal of 3 layers (concrete of the bridge, steel tube and the finally concrete in the tube) is consuming lot of time and represents heavy effort. Being so, we measured the bridge pre-stress by the use of MBN in one point only (on 5 different wires of the tendon). 

Manuscript: We prefer no change.

Reviewer: Figs. 4-8 could be jointed in two figures: 4(a) presents all rms dependence and 4(b) presents all PP dependence. It could be problem to show all operation (bridge) values in one graph, but it could look better.

Response: We tried to edit such figures, but these figures contain too much lines and curves.

Manuscript: We prefer no change due to lack of readability of the proposed figures.

Reviewer: How do you evaluate the field in kA/m for Rollscan measurements?

Response: We measured the magnetic field between the magnetising poles by the use of Hall probe in the preliminary phase of experiments for the magnetising conditions indicated in the study. The strength of magnetising field was measured in mT and converted into kA.m-1 afterward.

Manuscript: The field between the magnetising poles was measured by the use of the Hall probe in the preliminary phase of experiments for the magnetising conditions indicated in the study. The strength of magnetising field was measured in mT and converted into kA.m-1 afterward.

Reviewer: I do not like much the discussion about two anisotropy axes with the stress. There are more classical older works proposing dislocation formation as a reason for the following rms decrease. 

Response: We disagree with the dislocation formation as a reason for MBN decrease along with stress in this particular case. We agree that MBN decrease with stress in the plastic regime of loading when dislocation density increases as a result of microstructure hardening. However, in this particular case the wires loading is in elastic regime only since yield stress point is above 1300 MPa.  Being so, dislocation density is untouched and constant along with tensile stress loading.

We have already measured the evolution of MBN along with the stresses for the new wires in the elastic regime up to 1500 MPa and we have found that MBN decrease with stresses continuously and monotonically in the whole range of investigated stresses (from 0 up to 1500 MPa). These wires are in pre-stressed state during annealing and remarkable decrease of MBN after heat treatment is due to remarkable stress annealing anisotropy which completely consumes magnetocrystalline anisotropy (MBN increases with the tensile stress when magnetocrystalline anisotropy dominates only).

Manuscript: We prefer no change in the manuscript.

Reviewer 2 Report

see below. The article is clearly written.

Author Response

Reviewer: The article is clearly written.

Response: Thank you for appreciation of our work.

Manuscript: No change.

Reviewer 3 Report

The subject of the paper is interesting.

Please, consider these points:

The acronym ‘SEM’ could be defined on line 109. The quality of figure 1 could be improved Quality of equations 1 and 2 The font size of the paragraph on line 147 could be reduced The acronym ‘PP’ of line 227 should be italicized

Author Response

Reviewer: The acronym ‘SEM’ could be defined on line 109.

Response: We agree. We added definition of SEM.

Manuscript: “… by the use of Scanning Electron Microscopy (SEM) as well as …”

Reviewer: The quality of figure 1 could be improved

Response: We try to improve the contrast and brightness of Fig. 1.

Manuscript: Please, check appearance of Fig. 1.

Reviewer: Quality of equations 1 and 2

Response: We agree. The quality of the equations was really poor. We improved the quality of these equations.

Manuscript: Please, check appearance of equations 1 and 2 – page 11.

Reviewer: The font size of the paragraph on line 147 could be reduced.

Response: We checked the font size.

Manuscript: We checked the font size.

Reviewer: The acronym ‘PP’ of line 227 should be italicized 

Response: Thank you for this. We have made correction.

Manuscript: “…to the difference in MBN (PP) and their evolution …”
